

Natural Hazards and Earth System Sciences

EGU

Discussions

**Emergency management of the 2010 Mt. Rotolon landslide by**
**means of a local scale GB-InSAR monitoring system**
William Frodella[1], Teresa Salvatici[1], Veronica Pazzi[1], Stefano Morelli[1], Riccardo Fanti[1]
1. Department of Earth Sciences, University of Firenze, Via La Pira 4, 50121, Florence, Italy
*Correspondence to*: William Frodella (william.frodella@unifi.it)
**Abstract**
Diffuse and severe slope instabilities affected the whole Veneto region (Northeast Italy) between October 31st and
November 2nd 2010, following a period of heavy and persistent rainfall. In this context on November 4th 2010 a large
detrital mass detached from the cover of the Mt. Rotolon Deep Seated Gravitational Slope Deformation (DSGSD),
located in the upper Agno River Valley, channelizing within the Rotolon Creek riverbed and evolving into a highly
mobile debris flow. The latter phenomena damaged many hydraulic works, also putting at high risk bridges, local roads,
together with population of the Maltaure, Turcati and Parlati villages located along the creek banks and of the Recoaro
Terme town. Starting from the beginning of the emergency phase, the Civil Protection system was activated, involving
the National Civil Protection Department, Veneto Region, and local administrations personnel and technicians, as well
as scientific institutions. On December 8th 2010 a local scale monitoring system, based on a Ground Based
Interferometric Synthetic Aperture Radar (GB-InSAR), was implemented in order to evaluate the slope deformation
pattern evolution in correspondence of the debris flow detachment sector, with the final aim of assessing the landslide
residual risk and manage the emergency phase. This paper describes the outcomes of a two years GB-InSAR
monitoring campaign (December 2010 - December 2012), its application for monitoring, mapping, and emergency
management activities, in order to provide a rapid and easy communication of the results to the involved technicians
and civil protection personnel, for a better understanding of the landslide phenomena and decision making process in a
critical landslide scenario.

**1 Introduction**
Deep Seated Gravitational Slope Deformations (DSGSD) are normally not considered hazardous phenomena, due to
their typical very slow evolution, nevertheless under certain conditions ground movements can accelerate evolving into
faster mass movements or favoring collateral landslide processes (Crosta, 1996; Crosta and Agliardi, 2003). Therefore,
a multidisciplinary approach is fundamental in order to understand the complex nature of such phenomena, with the aim
of assessing the correct mitigation measures. In this framework advanced mapping methods, based on spaceborne,
aerial and terrestrial remote sensing platforms, represent the optimal solution for landslide detection, monitoring and
mapping, in different physiographic and land cover conditions, with special regards to large phenomena and hazardous
non accessible sectors (Casagli, 2017b; Guzzetti et al., 2012). In the last decades, many advanced remote sensing
technologies have been increasingly being recognized as efficient remote surveying techniques for the characterization,
and monitoring of landslide-affected areas, in terms of resolution, accuracy, data visualization, management, and
reproducibility, such as: digital photogrammetry (Chandler, 1999; Zhang et al., 2004), laser scanning (Abellan et al.,
2006; Gigli et al., 2012, 2014c; Jaboyedoff et al., 2012; Tapete et al., 2012), Infrared Thermography (Teza et al., 2012;





Gigli et al. 2014a, b; Frodella et al., 2015) and radar interferometry (Luzi et al., 2004; Bardi et al., 2014; Ciampalini et
al., 2016).
Ground Based Interferometric Synthetic Aperture Radar (GB-InSAR) systems in particular, for their capability of
measuring displacements with high geometric accuracy, temporal sampling frequency, and adaptability to specific
applications (Monserrat et al., 2014), represent powerful devices successfully employed in: a) engineering and
geological applications for detecting structural deformation, and surface ground displacements (Tarchi et al., 1997;
2003; Antonello et al., 2004; Casagli et al., 2010; 2017a), b) for the monitoring of volcanic activity (Nolesini et al.,
2013; Di Traglia et al., 2014a, b), and c) for analysing the stability of historical towns built on isolated hilltops (Luzi et
al., 2004; Frodella et al., 2016; Nolesini et al., 2016). Furthermore, in the recent years GB-InSAR technique has
developed to an extent where it can significantly contribute to the management of major technical and environmental
disasters (Del Ventisette et al., 2011; Broussolle et al., 2014; Lombardi et al., 2017; Bardi et al., 2017a, b). Between
October 31$^{st}$ 2010 and November 2$^{nd}$ 2010 the whole Veneto region territory (north-eastern Italy; Fig. 1) was hit by
heavy and persistent rainfall, that diffusely triggered floods and abundant slope failures, causing widespread damages to
people (3 fatalities and about 3500 evacuated people) and structures, furthermore resulting in heavy economic losses for
the agricultural, livestock, and industrial activities. In this context on November 4$^{th}$ 2010, part of detrital cover of the
Rotolon DSGSD suffered the detachment of a mass approximately 320000 m$^3$ in volume, that channelized in the
Rotolon Creek bed causing a large debris flow. This phenomenon was characterized by more than three kilometres of
run-out distance, damaging various hydraulic works and infrastructures (creek dams, weirs, bank protections), putting at
high risk the infrastructures (bridges, local roads, houses), together with the population of the inhabited areas located
nearby the creek banks (villages of Maltaure, Turcati, Parlati and the town of Recoaro Terme; Fig. 1). On December 8$^{th}$
2010 a GB-InSAR monitoring system was implemented in order to assess the landslide residual displacements and
support the local authorities for the emergency management (Fidolini et al., 2015). In this framework the Civil
Protection system was activated in order to manage the landslide emergency phase, by involving the national (DPC) and
regional (DPCR) Civil Protection Departments, in cooperation with scientific institutions (namely "Competence
centres", CdCs), local administration personnel, and technicians (Bertolaso et al., 2009; Pagliara et al., 2014;
Ciampalini et al., 2015). Accurate geomorphological field surveys were also carried out in this phase, in order to
analyse the landslide morphological features as to improve the radar data interpretation (Frodella et al., 2014; 2015;
2017). In addition a landslide 3D runout numerical modelling was performed with the aim of identifying possible debris
flow events source and impacted areas, flow velocity and deposit distribution within the Rotolon creek valley (Salvatici
et al., 2017). This paper is focused on the outcomes of a long-term continuous GB-InSAR monitoring campaign
(December 2010 - December 2012) carried out during the post-event recovery phase, in which monitoring, mapping,
and emergency management activities were implemented for assessing the landslide residual risk and analyse its
kinematics. In this framework field activities were carried out by local Civil Protection operators and technicians for a
validation of the remotely sensed data (landslide area inspections). In particular, the analysed radar data were shared
with the involved technicians and civil protection personnel in order to provide a rapid and easy communication of the
results, and enhance the synergy with all of the subjects involved in the recovery phase.
**2. Study area**
The Rotolon DSGSD is located in the Vicentine Prealps, on the south-eastern flank of the Little Dolomites chain, in the
uppermost Agno river valley (Fig. 1). The instability processes of the valley, such as slope failures and debris flows


induced as secondary phenomena of the DSGSD, have threatened the Upper Agno valley for centuries (Frodella et al.,

77   2014).

From a geologic point of view the landslide develops in the uppermost portion of a sub-horizontally bedded mainly
dolomitic-limestone stratigraphic succession, from middle Triassic to lower Giurassic in age, belonging to the South
Alpine Domain (De Zanche and Mietto, 1981).


**Figure 1.** Geological sketch map of the Upper Agno River Valley with the location of the Rotolon landslide.
The mass movement is delimited to the NW by the ridge of the Mount Obante group and develops from about 1700 to
1100 m a.s.l., covering an area of 448000 m$^2$. The Rotolon DSGSD can be classified as a DSGSD ("Sackung type";
Zischinsky, 1969), and it is characterized by a complex activity (Cruden and Varnes, 1996) that leads to a rough
physiographic, characterized by steep scarps, trenches, crests and counterscarps (Figs. 2 and 3). Two distinct sectors can
be identified, basing on the acting dominant slope instability processes: i) an upper "Detachment sector", followed
downstream by a ii) "Dismantling sector" (Frodella et al., 2014). The Detachment sector (having a mean slope of 30°),
develops downstream of the main landslide crown (Figs. 2a and b; Fig. 3), and it's dominated by extensional
deformation that leads to the development of tensional fractures, resulting in alternate trenches and crests which creates
very rough, stepped topographic surface. This area is affected both by gravitational and erosional processes and by the
rock mass detensioning and disaggregation, which cause the accumulation of various depositional elements (colluvial
fans, colluvial aprons, rock fall and rock avalanche deposits) formed by very coarse and heterometric clasts, ranging




from cobbles to boulders with scattered blocks (decimetric to decametric in size) in a coarse sandy matrix (Figs. 3 and

96   4).

The Dismantling sector (mean slope of 34°) includes sectors formed by sub-vertical highly weathered rock walls. It is
dominated by surficial processes (e.g., concentrated and diffuse erosion, slope-waste deposition due to gravity, detrital
cover failures) that widely cover the evidences of deeper deformations (Figs. 3 and 4). The Dismantling sector supplies
material for debris flows, which channelize downstream within the Rotolon Creek bed, therefore representing the most
critical sector with respect to short-term hazardous phenomena.

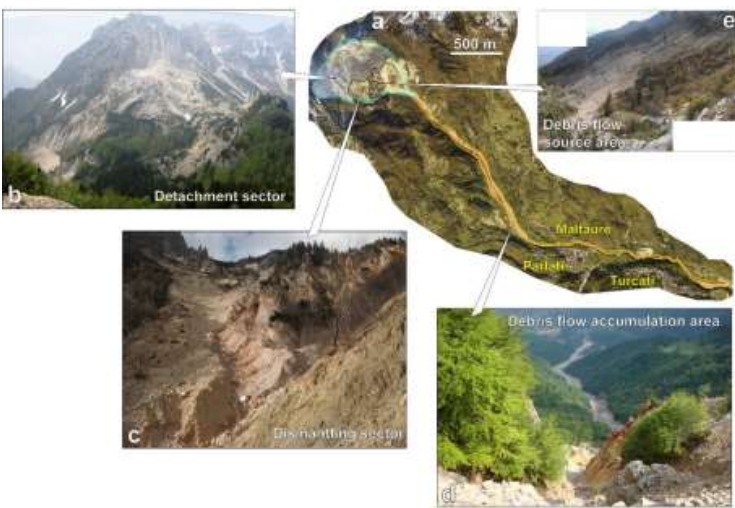


**Figure 2.** The Mt. Rotolon DSGSD plan **(a)**; landslide sectors **(b, c)** and the 2010 debris flow features **(d, e)**.



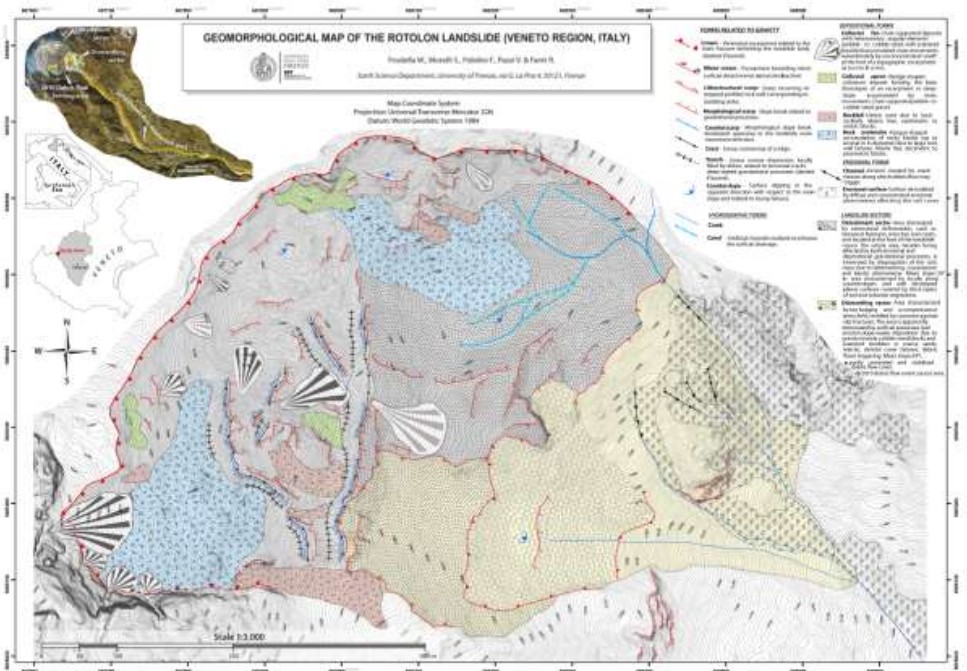

**Figure 3.** Geomorphological map of the Rotolon Landslide (modified after Frodella et al., 2014).

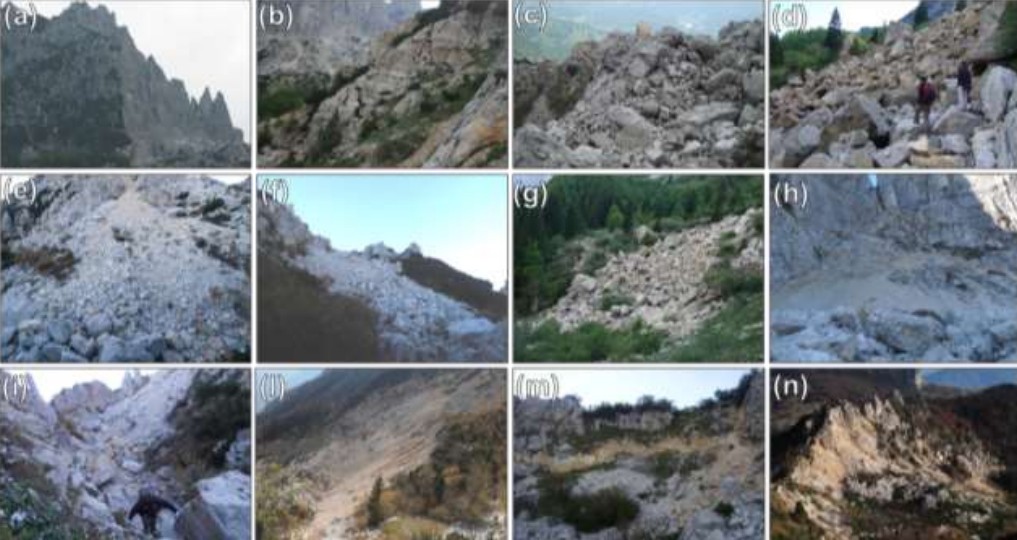

**Figure 4.** Geomorphic and sedimentary features of the Mt. Rotolon DSGSD. Detachment sector: **(a)** rock walls prone to
rock falls; **(b, c)** rock mass affected by different stages of disaggregation; **(d)** plurimetric rock blocks within rock
avalanche deposit. Main depositional elements within the landslide body: **(e)** colluvial fan; **(f, g)** channelized and
diffused rock fall deposits; **(h)** colluvial aprons. Main landslide linear elements: **(i)** landslide trench; **(l)** 2010 debris
flow detachment scarp; **(m)** DSGSD crown sector; **(n)** landslide crest.





### 3. The GB-InSAR technique basic theoretical principles

The GB-InSAR is a computer-controlled microwave transmitting and receiving antenna, that moves along a mechanical linear rail in order to synthesize a linear aperture along the azimuth direction (Tarchi et al., 1997). It radiates an area with microwaves in the Ku band (12-18 GHz) and registers the backscattered signal in the acquiring time interval (from few to less than 1 minute with the most modern systems): each acquisition produces a complex matrix of values from which phase and amplitude information are calculated (Luzi et al., 2004; Luzi, 2010). A SAR image contains amplitude and phase information of the observed objects backscattered echo within the investigated scenario, and it is obtained by combining the spatial resolution along the direction perpendicular to the rail (range resolution, $\Delta Rr$) and the one parallel to the synthetic aperture (azimuth or cross-range resolution, $\Delta Raz$) (Luzi, 2010). The working principle of the GB-InSAR technique is the evaluation of the phase difference, pixel by pixel, between two pairs of averaged sequential SAR complex images, which constitutes an interferogram (Bamler and Hartl, 1998). The latter does not contain topographic information, given the antennas fixed position during different scans (zero baseline condition). Therefore, in the elapsed time between the acquisition of two or more subsequent coherent SAR images, it is possible to derive from the obtained interferograms a 2D map of the displacements that occurred along the sensor LOS (with a millimeter accuracy in the Ku band) (Tarchi et al., 1997; 2003; Pieraccini et al., 2000; 2002). The capability of InSAR to detect ground displacement depends on the persistence of phase coherence (ranging from 0 to 1) over appropriate time intervals (Luzi, 2010). Among the technique's advantages it should be highlighted that GB-InSAR works: a) without any physical contact with the slope, avoiding the need of accessing the area; b) in almost every light and atmospheric condition; c) continuously over a long time; d) with a millimetre accuracy; e) providing near real time detailed and spatially extensive information.

This latter feature in particular gives a strong advantage with respect to traditional ground surface methods (like inclinometers, extensometers, total stations), which on the contrary provide single-point information, generally are not sufficient to evaluate the kinematic and behaviour of complex landslide. The main drawback of the technique is the logistics of the installation platform, both because the GB-InSAR system measures only the displacement component parallel to the line of sight (L.O.S.), and because the azimuth resolution (the ability to separate two objects perpendicular to the distance between the sensor and the target) reduces with the increase of the distance with respect to the target (Fig. 5). Moreover, vegetated areas can be another drawback of the technique since they are commonly characterized by low coherence and power intensity.

### 4. The adopted monitoring system

The GB-InSAR system was installed in the Maltaure village, at an average distance of 3 km with respect to the landslide, pointing upwards to NW (Fig. 5). The radar parameters are summarized in Fig. 5. Given the acquisition setting of the site and the civil protection purposes, the radar data covers an area of 1.2 km$^2$. The logistics of the GB-InSAR system installation favored a good spatial coverage of the data on the monitored area, especially with special regards to the Dismantling sector. Nevertheless, shadowing effects, due to the slope roughness, crests and counter-slope surfaces affect the Detachment sectors (Figs. 5 and 7).



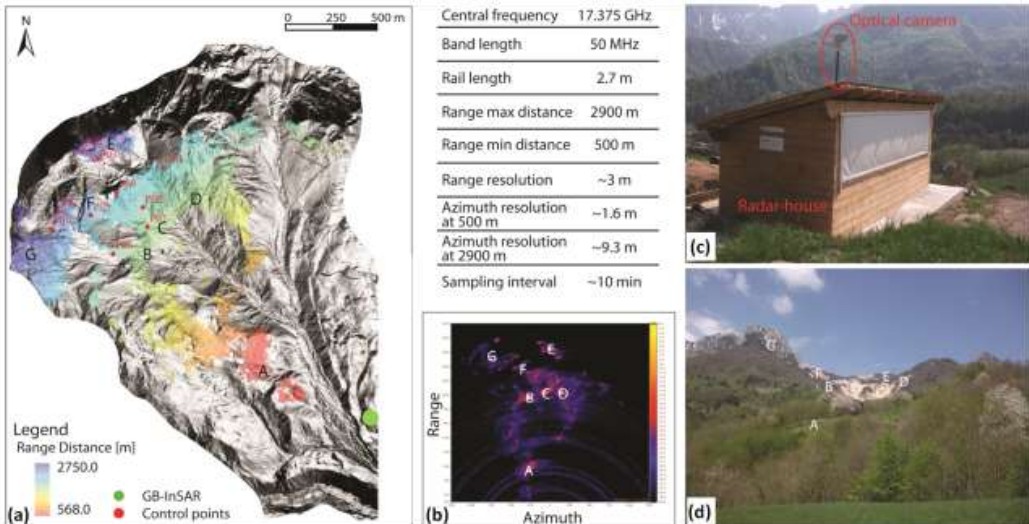


**Figure 5.** The adopted monitoring system: **(a)** Location of the GB-InSAR system and radar data coverage features (A-
G=recognized landslide sectors); **(b)** the adopted monitoring parameters and radar power image, displaying the
correspondent recognized landslide sectors; **(c)** the radar system hut setting; **(d)** picture of the monitoring optical system
scenario (A-G=corresponding sectors).
The radar system acquired GB-InSAR data every 10 minutes, from which cumulated 2D displacement maps, and
displacements time series of 10 measuring points (Fig. 5) were obtained. GB-InSAR data were processed using
LiSALab software (Ellegi s.r.l.) and uploaded via LAN network: i) on a dedicated Web-based interface, allowing for a
near real time data on-routine visualization; ii) on a remote ftp server (in ASCII format), in order to perform on demand
analysis in case of critical weather events based on the national civil protection weather forecast system (Fig. 6).
The latter were performed integrating into a GIS environment the displacement maps and comparing them with
ancillary data (rainfall, geological and geomorphological maps). In addition, a remotely adjustable robotized high
resolution optical camera (Ulisse Compact model produced by Videotec S.p.A, digital zoom 10x - 36x) manageable via
IP-Ethernet interface was installed in correspondence of the radar system, acquiring data every 60 minutes and allowing
for programmable zooms, with the aim of checking of the landslide hazardous and inaccessible Dismantling sector
(Figs. 5 and 6). The time line rationale of the monitoring system and emergency management procedures is summarized
in Fig. 6.





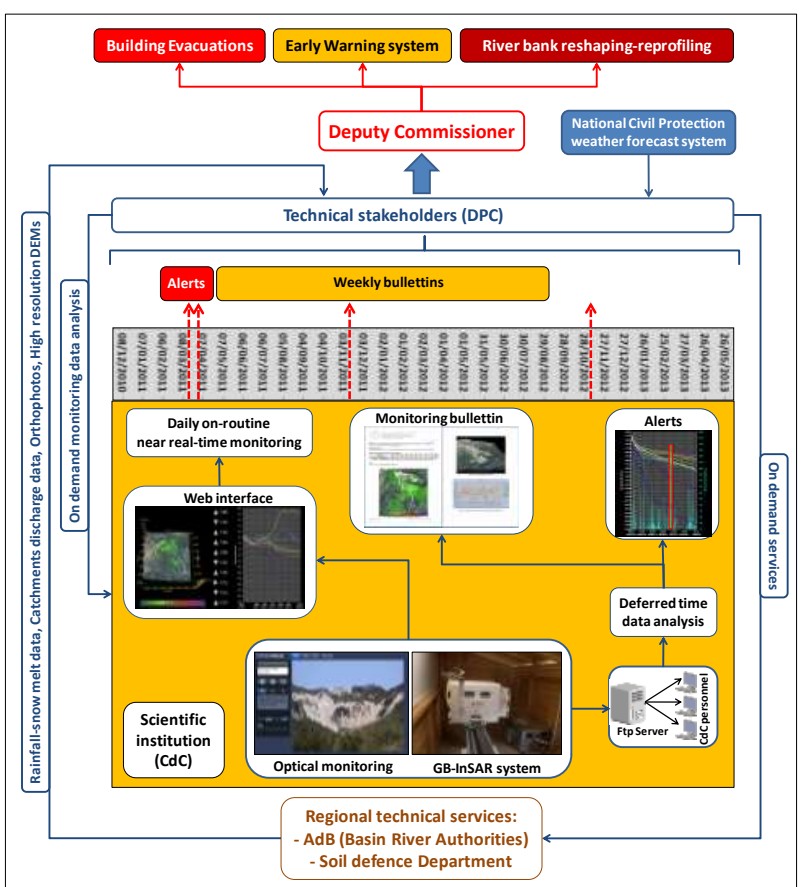

**Figure 6.** Time line rationale of the Rotolon monitoring system and emergency management procedures.
**5. GB-InSAR data analysis**
The obtained GB-InSAR incremental cumulative displacement (ICD) maps and measuring points displacement time
series are shown in Figs. 7 and 8, respectively. By using a selected colour scale, the obtained radar maps are displayed
as a function of the displacement measured in the period covered by the acquisitions, spanning from December 8[th] 2010
up to the beginning of each month of the monitoring campaign, until the end of the monitoring period (the negative
displacement values indicate movements approaching to the sensor; Fig. 7). In order to evaluate the deformation rates
and provide an easy-interpretable data, a traffic light type colour scale was applied in all the displacement maps.
GB-InSAR measuring points (corresponding to a 5 x 5 pixel size area) were selected in correspondence with sectors
where the radar signal is characterized by high stability, in order to monitor the landslide kinematics and characterize
the various landslide physiographic features (Fig. 7). Furthermore with the aim of performing a temporally detailed
displacement analysis and detecting the spatial pattern of residual landslide deformation, monthly cumulated
displacement (MCD) maps were also selected and analysed from the collected GB-InSAR dataset (Fig. 9).


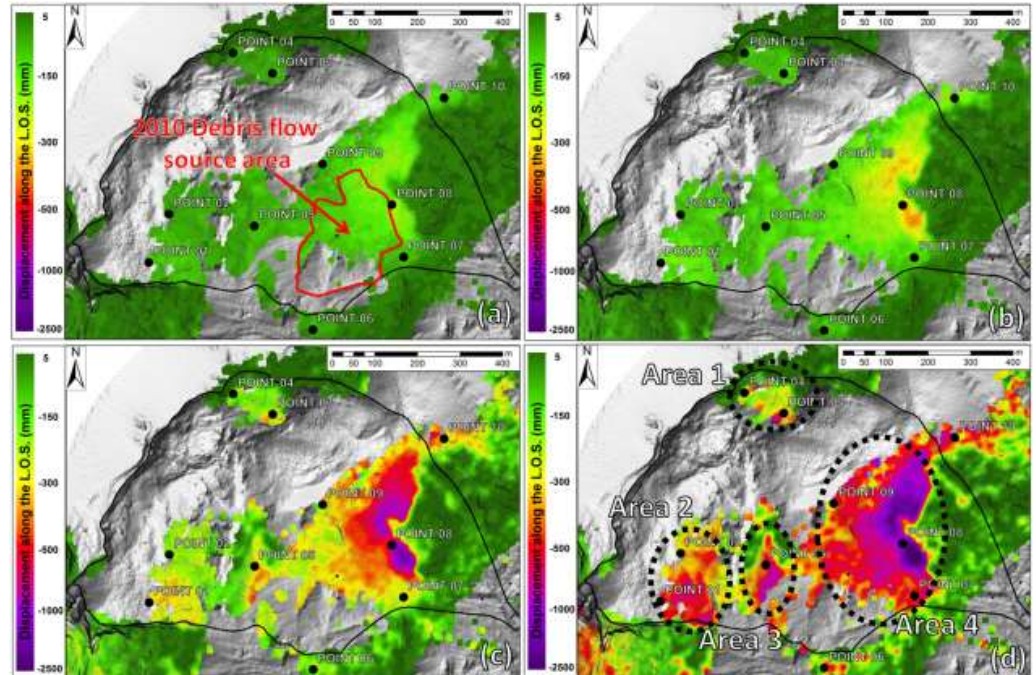


**Figure 7.** ICD maps of the Rotolon landslide: (a) December 8th 2010 - January 1st 2011; (b) December 8th 2010 - February 1st 2011; (c) December 8th 2010 - December 1st 2011; (d) December 8th 2010 - December 31th 2012 (Point 1-10 represent the GB-InSAR measurement points in correspondence of which the displacement time series were extracted).

From the analysis of the collected GB-InSAR dataset of the ICD maps (Fig. 7) four distinct areas characterized by relevant residual cumulated displacement were identified (Fig. 7d):

- Area 1 (ICD=737 mm, about 12500 m² in extension) and Area 2 (ICD=751 mm, area of 28000 m²), corresponding to the material infilling the Detachment sector (Fig. 2), such as minor rock fall and rock avalanche deposits;

- Area 3 (ICD=960 mm; 12000 m² in extension) and Area 4 (ICD=2437 mm; 88000 m² coverage), both falling within the Dismantling sector detrital cover (Fig. 2) which was not affected by the 2010 debris flow detachment.

The measuring points time series (Fig. 8) display cumulated displacements ranging from 337 mm (Point 6) to 595 mm (Point 4, located in Area 1); Point 8 in particular (falling within Area 4) displays the monitored area cumulated peak displacements (ICD=1476 mm), showing two acceleration periods (middle March 2011 and beginning of November 2011), alternating with a more linear trend. The comparison amongst the MCD maps highlighted a first phase of widespread residual displacements (December 2010, Fig. 9a), which gradually decreased starting from the following month (Fig. 9b). In the following period ground deformation took place in correspondence of limited sectors within Area 4 (May 2011 in particular shows the higher MCD up to 244 mm; Fig. 9d), except for a widespread reactivation recorded in November 2011 (Fig. 9e).

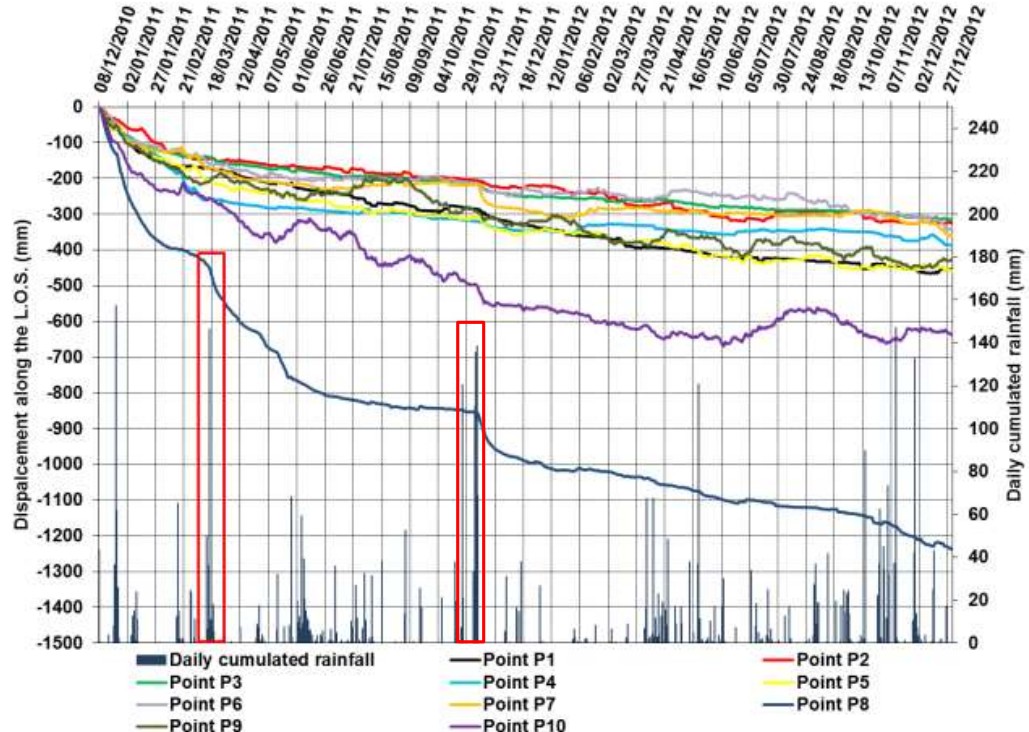

198

**Figure 8.** Selected measuring points displacement time series of the monitored scenario (red squares enhance Point 8
accelerations).

Furthermore, in order to automatically extract the most hazardous residual displacement sectors, the MCD dataset was
analysed by means of a MATLAB code (Salvatici et al., 2017) (Fig. 10). The code extracts from the dataset all of the
areas affected by deformation higher than a selected threshold value, set equal to 92.3 mm, being the minimum
displacement among all the maximum MCD values. The results are displacement maps showing only the areas with
such selected displacements (Fig.10 a-d), confirming the trend highlighted by the MCD maps (Fig. 9). The second
operation of the employed code consists in the frequency calculation of displacement occurred (the code computes how
many times each pixel has recorded the selected displacement during the monitoring period) (Fig. 10e). By using this
method, three critical areas characterized by repeated residual reactivations were detected: Area 2, Area 3 (1
reactivation) and especially Area 4 (8 reactivations).


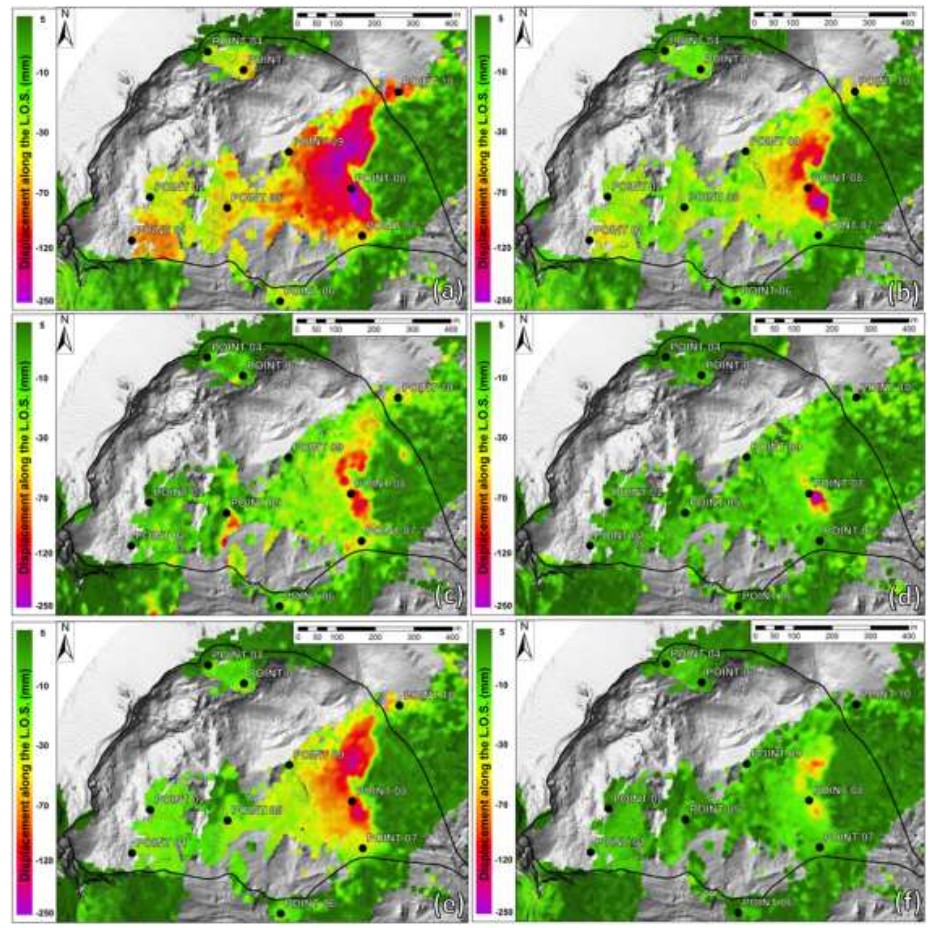

210

**Figure 9.** Selection of MCD maps from the GB-InSAR dataset: (a) December 2010 (232 mm cumulated peak
displacement); (b) January 2011 (214 mm); (c) March 2011 (173 mm); (d) May 2011 (244 mm); (e) November 2011
(174 mm); (f) November 2012 (106 mm).

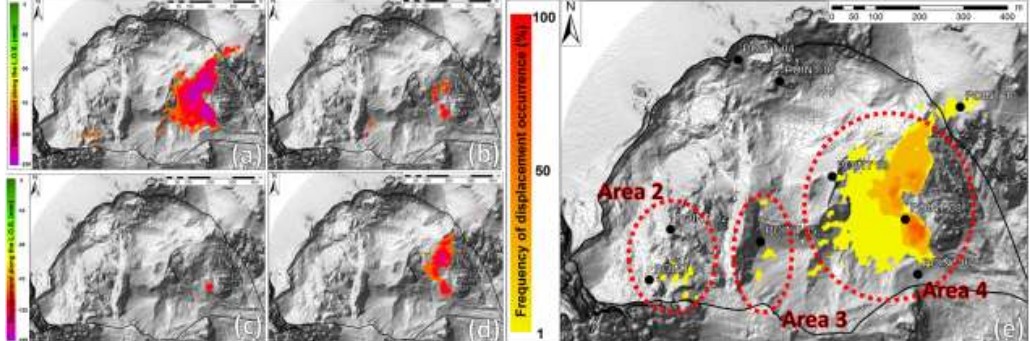

214

**Figure 10.** Residual reactivation maps obtained from selected MCD maps by means of the employed MATLAB code
analysis: a) December 2010; (b) March 2011; (c) May 2011; (d) November 2011; (e) frequency map of the reactivation
of the critical residual displacement sectors, classified basing on their activation frequency.





## 6. Discussion

Successful strategies for landslide residual hazard assessment and risk reduction would imply integrated methodology for instability detection, mapping, monitoring, and forecasting (Confuorto et al., 2017). In order to provide information on the nature, extent and activation frequency of ancient landslides, standard detection and mapping procedures need a combination of field-based studies and advanced techniques, such as remote sensing data analysis and geophysical investigations (Ciampalini et al., 2015; Lotti et al., 2015; Del Soldato et al., 2016; Morelli et al., 2017; Pazzi et al., 2017a, b). In particular GB-InSAR represents a versatile and flexible technology, allowing for rapid changes in the type of data acquisition (geometry and temporal sampling) based on the characteristics of the monitored slope failure, which is capable of assessing the extent and the magnitude of the landslide residual hazard (Di Traglia et al., 2014; 2015; Carlà et al., 2016a, b). In the presented case study the 2 year continuous GB-InSAR monitoring campaign allowed to measure the slope displacement with a millimetre accuracy over a 1.2 km square landslide area, enabling to analyse the evolution pattern of the landslide residual hazard. By comparing the landslide geomorphological map (Frodella et al., 2014) with the ICD displacement map of whole monitored period (Fig. 11), the four critical areas shown in Figure 7 are analysed in detail:

- Area 1, including measuring Points 3 and 4, is located in the northern side of Detachment sector (Fig. 11a). The points recorded in the first few months (between December 2010 and March 2011), a peak of displacements of about 260 mm (Point 4) and 150 mm (Point 3), after this period the displacement decreased up to 8th November 2011. Between 8th and 12th November, during a major rainfall event (68 mm), the displacements increased again (Fig. 11b). The displacements recorded by the points within Area 1 may are related to deformations affecting the deposits placed along the steep scarp connected to the main crown delimiting the DSGSD (Fig. 4).

- Area 2 is located in the Detachment sector (SW side of the DSGSD). Two measuring points (Points 1 and 2) therein located (Fig. 11d) recorded a peak of displacement between December 2010 and March 2011 of about 170 mm (Point 1) and 130 mm (Point 2), respectively. The ground deformations recorded by these points are related to slope waste deposition due to gravity of the coarse material infilling this sector, such as ancient rock avalanche deposits (Point 1) and detensioned rock mass (Point 2) (Fig. 4).

- Area 3 represents the border between Detachment and Dismantling sectors, located upstream the 2010 event scarp (Fig.11c). Its kinematics is represented by Point 5 behaviour, which may be associated with sliding of the material infilling the materials, showing a similar trend with respect to P1 (Figs. 4-11d).

- Area 4 represents the lowermost portion of Dismantling sector. Three measuring points are therein located: Points 7, 8 and 9 (Figs. 11e and 11f). Points 7 and 8 display the kinematics the detrital cover surrounding the 2010 debris flow triggering area. Both control points show acceleration periods alternating with of periods of stability. In particular the trend of P8, located near the Rotolon creek ephemeral springs and channels (Frodella et al., 2014; 2015) shows a correlation with cumulative precipitation above a threshold value of about 100 mm (Fig. 11f), which contribute to the sub-surface water circulation within the detachment sector loose detrital cover. This suggests that the recorded displacements may be associated to the spring erosion within the detrital cover. This point records the maximum displacement of the entire area monitored by GB-InSAR system of about ICD=1236 mm. The area it is apparently dominated by superficial processes, such as widespread soil erosion and slope-waste deposition due to gravity. Measuring Point 9, located nearby the Dismantling sector upstream limit, records cumulative displacement of 445 mm and shows an irregular trend mainly due to its location near vegetated areas (Figs. 4-9f).



257

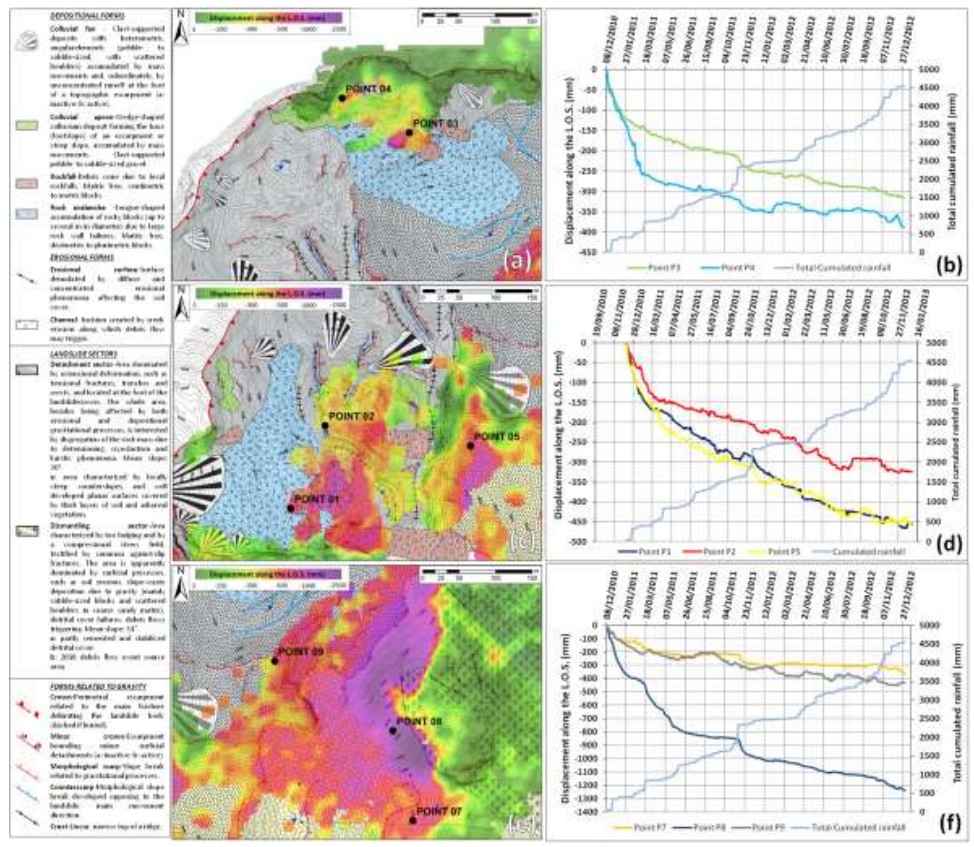

258

**Figure 11.** Integration between geomorphological map (modified after Frodella et al., 2014), the ICD maps displacement maps of whole monitored period, and the control points displacements time series: (a) the zoom of the Area 1 shown in Figure 7d; (b) the displacements time series of P4 and P3; (c) zoom of the Area 2 and Area 3 shown in Figure 7d; (d) the displacement time series of Points 1, 2 and 5; (e) zoom of Area 4 shown in Figure 7d; (f) the displacement time series of Points 7, 8 and 9.

The use of GB-InSAR ICD maps and the integration with geomorphological field surveys proved its usefulness in recognizing Area 4 (located within the DSGSD Dismantling sector; Figs. 3 and 7) as the most hazardous sector within the monitored scenario, due to the widespread and intense recorded cumulated displacements (2437 mm), and its geomorphological features (steep slope, loose very coarse debris and acting surface widespread erosional processes due to the presence of ephemeral springs), and frequency of reactivations (Fig. 10).The main triggering factor for this area shallow remobilizations are intense rainfall events, as highlighted by measuring point 8 time series (Fig. 8). Area 3 (recording 960 mm of total cumulated displacements) falls as well within the Dismantling sector detrital cover, and was considered the second most hazardous landslide sector within the monitored scenario. Other areas characterized by relevant residual cumulated displacement were identified in Area 1 (737 mm) and Area 2 (751 mm), corresponding to the material infilling the Detachment sector (Fig. 2), but they were not considered hazardous due to a 300 meter long





and 20 meter high N-S trending trench acting as a physical barrier separating the upper Detachment sector from the
lowermost Dismantling sector.
Furthermore the comparison amongst the MCD maps (Fig. 9 and 10) highlighted widespread and frequent residual
displacements taking place in correspondence of Area 4 during the wet winter-fall months (December 2010=232 mm;
January 2011=214 mm; March 2011=173 mm; November 2011, 2012= 174 and 106 mm respectively). Nevertheless, in
May 2011 Area 4 reached the higher MCD in the monitored period (244 mm), although concentrated in a limited sector
located nearby the measuring Point 8 (Fig. 9d).
A simplified local scale early warning system (Intrieri et al., 2012) was implemented based on three different warning
levels: ordinary, pre-alarm and alarm levels (Figure 5). In order to ensure the safety for the post-recovery management
personnel, hourly displacement thresholds were adopted: the level change occurred if the following thresholds were
surpassed (<1 mm/h ordinary; between 1.0 mm/h and 5.0 mm/h for the pre-alarm; >5 mm/h for the alarm level).
Communication, which is a fundamental issue of every early warning system (Intrieri et al. 2013), was operated through
the dispatch of monitoring bulletins every week and whenever the warning thresholds were exceeded. In this
framework, based on the surface of the deformation areas and the increasing trends of displacement time series, 4
monitoring alerts were obtained: i) March 19th 2011; ii) April 7th 2011; iii) 8-12th November 2011; iv) November 10-
12th 2012 (Fig. 6). All these events were located in the area monitored by measuring Point 8. Inspections carried out by
the optical monitoring device and by means of field surveys from safe viewing points, assessed that detected
accelerations did not generate significant slope failures, although rainfall comparable to that of November 2010 had hit
the area. Following the second alert a weekly bulletin phase (May 2011 - September 2012) was planned for a residual
risk prevention strategy.

**7. Conclusions**
In the framework of the 2010 hazardous events affecting the Rotolon creek valley, a local scale GB-InSAR system was
implemented for mapping and monitoring slope landslide residual deformations and for early warning purposes in case
of landslide reactivations, with the aim of assuring the safety of the valley inhabitants and the personnel involved in the
post-event recovery phase. The radar system acquired GB-InSAR data every 10 minutes, from which cumulated 2D
displacement maps, and displacements time series of 10 measuring points were obtained. The analysed GB-InSAR data
were uploaded both on a dedicated Web-based interface and remote ftp server, allowing for a daily near real time data
on-routine visualization and on demand analysis in case of critical weather events. In this framework, based on the
surface of the deformation areas and the increasing trends of displacement time series, 4 monitoring alerts were
obtained and a 16 months weekly monitoring bulletin campaign was performed (May 2011-September 2012). All of the
monitoring data were shared with the technical stakeholders and decision makers involved in the emergency
management. The adopted monitoring system provided all of the technical personnel and the local authorities decision
makers involved in the post-crisis management activities with a reliable, rapid and easy communication system of the
monitoring results, designed in favour of an enhanced understanding of such a critical landslide scenario and an
improvement of decision making process. Based on the recorded residual deformations four critical sectors were
identified in the monitored scenario, on the basis of the measured cumulated displacements, frequency of activation and
geomorfological features. Amongst these sectors Area 3 and in particular Area 4 (recording respectively 960 mm and
2437 mm of total cumulated displacements) were considered the most hazardous for potential debris flow reactivations.
The latter areas are in fact located within a steep landslide sector characterized by loose detrital cover, affected by soil
erosion and slope-waste deposition (Dismantling sector). The displacement time series of the GB-InSAR measuring





points provided information on the landslide kinematics: displacements range from 337 mm (Point 6) to 1476 mm
(Point 8).
This latter displays the monitored area cumulated peak displacements, showing two acceleration periods (middle March
2011 and beginning of November 2011) triggered by intense precipitations, alternating with a more linear trend. The
kinematics of the other representative measuring points, is related either to deformations affecting the deposits placed
along the steep scarp connected to the main DSGSD (Points 3-4), or to slope waste deposition due to gravity affecting
the coarse material infilling the Detachment sector (Points 1-2-5). The comparison amongst the MCD maps highlighted
a first phase of widespread residual displacements (December 2010). In the following period, ground deformation took
place in correspondence of limited sectors within Area 4, except for a widespread reactivation recorded in November
2011. The acquired radar data suggest a complex nature of the monitored landslide: its geomorphological features (e.g.,
rough topography, stepped profile in its upper sector, showing scarps, counterscarps, ridges, trenches and counterslopes,
toe bulging) documents the activity of long-term deep seated processes, while the radar data recorded the wide spectrum
of short-term secondary instability phenomena, in terms related to erosional-depositional gravitational processes
(Detachment sector) and soil erosion/slope-waste deposition (Dismantling sector). Although this latter sector represent
the most hazardous area phenomena, the displacements therein acting in the analysed time span appear to be related to
ephemeral spring erosion within the loose detrital cover. This suggests that these processes are only the surficial and
secondary expression of a more complex deep seated landslide system. The here presented methodology could represent
a useful contribution for a better understanding of landslide phenomena and decision making process during the post-
emergency management activities in a critical landslide scenario (a populated mountainous area particularly devoted to
touristic activities). Furthermore, the methodology could be profitably adapted, modified, and updated in other
geological contexts.

**Acknoledgements**
The GB-InSAR apparatus used in this application was designed and produced by Ellegi s.r.l. and based on the
proprietary technology GB-InSAR LiSALAB derived from the evolution and improvement of LiSA technology
licensed by the Ispra Joint Research Centre of the European Commission. We also would like to thank the Veneto Soil
Defence Regional Direction for providing Lidar and aerial photo data.

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
