# Peer review of "GB-InSAR monitoring of slope deformations in a mountainous"

_Natural Hazards and Earth System Sciences, 2017_

## Referee Comment (RC1) · Anonymous Referee #1 · 24 Jul 2017

General comments:

The paper entitled "Emergency management of the 2010 Mt. Rotolon landslide by means of a local scale GB-InSAR monitoring system" deals with use of at local scale monitoring system based on Ground Based Interferometric Synthetic Aperture Radar (GB-InSAR) to assess a slope deformation pattern evolution in correspondence of a debris flow detachment sector, with the final aim to monitoring, mapping and evaluate the residual risk and manage the emergency phase. The title and content of the paper are according to the scope of the journal and generally the whole structure of the work is a well-written, highlighting a very interesting use of the innovative GB-InSAR monitor-

ing technique that, integrated with in situ field survey and investigations, provide useful information in order to better understand the "slow or very slow-landslides phenomena" as the Deep Seated Gravitational Slope Deformation (DSGSD) exanimate in this case study as well as in decision making processes and emergency management activities. For my opinion, the paper should be considered for publication after a minor revision (following the Specific comments provided below) which could give an improvement to the work.

Specific comments:

In order to improve the manuscript, the following points should be considered by the authors while revising the paper. In particular:

- in the Introduction section, when the Authors speaking about the use of innovative technologies for the characterization and monitoring of landslide-affected areas (see from line 32 to line 38), including remote sensing techniques and radar inteferometry (both terrestrial and satellite), in according to scientific literature, the authors should include some other references such as: Gullà et al., 2017; Peduto et al., 2017a; Tofani et al., 2014.

- in the section 3, at the line 131, the Authors speak in general about of a millimeter accuracy of the acquired data by GB-InSAR. Give more detail about the real accuracy (range values). A comparison with conventional ground monitoring techniques, was carried out? What are the differences on the accuracy also compared with the InSAR data provided by satellite sensors? It might be useful to provide a comparison whit the values included in the works of Nicodemo et al., 2016; Peduto 2017b; Casu et al., 2006) about the accuracy on the average velocities or displacements data derived by satellite radar sensors processed by InSAR or DInSAR techniques.

- in the section 5 as well as in the figures 7,9 and 10, the Authors refer to incremental cumulative displacement (ICD) or monthly cumulated displacement (MCD) evaluated along the LOS direction. Why not along the real movement direction? Could be per-
formed a data projection? Please, provide further details about this.

- for a better understanding, an improvement of the Figures 3 and 11 is necessary. In particular, a visible legend should be provided.

References:

Casu, F., Manzo, M., Lanari, R. (2006). A quantitative assessment of the SBAS algorithm performance for surface deformation retrieval from DInSAR data. Remote Sensing of Environment., 102 (3-4), 195-210.

Gullà, G., Peduto, D., Borrelli, L., Antronico, L., Fornaro, G. (2017). Geometric and kinematic characterization of landslides affecting urban areas: the Lungro case study (Calabria, Southern Italy). Landslides 14:171–188. doi:10.1007/s10346-015-0676-0.

Nicodemo G, Peduto D, Ferlisi S, Maccabiani J. (2016). Investigating building settlements via very high resolution SAR sensors. In: Bakker J, Frangopol D.M., van Breugel K (eds) © 2017Life-cycle of engineering systems: emphasis on sustainable Civil Infrastructure. Taylor & Francis Group, London, pp 2256–2263.

Peduto D., Ferlisi S., Nicodemo G., Reale D., Pisciotta G., Gullà G. (2017a). Empirical fragility and vulnerability curves for buildings exposed to slow-moving landslides at medium and large scales, Landslides, in press, DOI : 10.1007/s10346-017-0826-7.

Peduto, D, Nicodemo, G, Maccabiani, J, Ferlisi, S. (2017b). Multi-scale analysis of settlement induced building damage using damage surveys and DInSAR data: a case study in The Netherlands. Engineering Geology, 218:117–133. doi: 10.1016/j.enggeo.2016.12.018.

Tofani, V., Raspini, F., Catani, F., Casagli, N., 2014. Persistent scatterer interferometry (PSI) technique for landslide characterization and monitoring. In: Sassa, K., Canuti, P., Yueping, Y. (Eds.), Landslide Science for a Safer GeoenvironmentMethods of Landslide Studies 2. Springer International Publishing, pp. 351–357 (ISBN: 9783319050492).

---

## Referee Comment (RC2) · Anonymous Referee #2 · 28 Jul 2017

The English in which this paper is written is awkward in places, although comprehensible, and the authors should use shorter paragraphs.

The title of the paper is misconceived. It is not a study of emergency management but a description of the deployment and use of a landslide monitoring and alarm system.

Line 9: Deep Seated Gravitational Slope Deformation - as it is not a common noun and adjectives, this term should not be capitalised. The same issue occurs with other terminology.

Line 33: have been increasingly being recognized - adjust, please

Line 79: Jurassic

Line 90: please do not use contractions in formal prose.

Figure 3 includes wording that will not reproduce at the page scale. it should be re-drafted.

Overall, I think the paper is publishable with a few tweaks to the English and a revised title. It offers a good description of an innovative and valuable landslide monitoring system.

---

## Author Comment (AC1) · 3 Aug 2017

Dear Referee,

We would like to thank you for your encouraging comments, which we largely agree upon. We are sure that the manuscript will greatly benefit from your suggestions. Hereafter the list of your comments is reported, followed by our response. We will also provide a version of the manuscript with the tracked revisions.

Referee #1 comments: in the Introduction section, when the Authors speaking about the use of innovative technologies for the characterization and monitoring of landslide-

affected areas (see from line 32 to line 38), including remote sensing techniques and radar interferometry (both terrestrial and satellite), in according to scientific literature, the authors should include some other references such as: Gullà et al., 2017; Peduto et al., 2017a; Tofani et al., 2014.

Authors: the proposed references were included.

Referee #1 - in the section 3, at the line 131, the Authors speak in general about of a millimeter accuracy of the acquired data by GB-InSAR. Give more detail about the real accuracy (range values). A comparison with conventional ground monitoring techniques, was carried out? What are the differences on the accuracy also compared with the InSAR data provided by satellite sensors? It might be useful to provide a comparison whit the values included in the works of Nicodemo et al., 2016; Peduto 2017b; Casu et al., 2006) about the accuracy on the average velocities or displacements data derived by satellite radar sensors processed by InSAR or DInSAR techniques.

Authors: More details about the range value accuracy were given in the text (including the suggested works). References were also given in the discussion section about an automated total station working in the landslide area in during our research (see the manuscript revised version).

Referee #1 - in the section 5 as well as in the figures 7,9 and 10, the Authors refer to incremental cumulative displacement (ICD) or monthly cumulated displacement (MCD) evaluated along the LOS direction. Why not along the real movement direction? Could be performed a data projection? Please, provide further details about this.

Authors: It is well known that a GB-InSAR system is able to measure only the component of the movement parallel to the LOS of the instrument. Thus the real displacement vector of the observed object can be calculated only if its direction is a priori known. This is one of the major limits of the technique. This is why usually the instrument is set with the view direction as parallel as possible to the expected deformations. The current paper was centered on the application of a monitoring system applied to a particular case study (a debris-flow affected slope in a mountainous inhabited area), which results could be shared with the involved technical personnel. Therefore we focused on easy interpretable data, while the data projection on the slope in order to obtain of the real movement direction will be the objective of a future work.

Referee #1 - for a better understanding, an improvement of the Figures 3 and 11 is necessary. In particular, a visible legend should be provided.

Authors: The legend of Figure 3 and 11 were improved.

Please also note the supplement to this comment:
https://www.nat-hazards-earth-syst-sci-discuss.net/nhess-2017-207/nhess-2017-207-AC1-supplement.pdf

**Supplement:**

Dear Revisors,

We would like to thank you for your encouraging comments, which we largely agree upon. We are sure that the manuscript will greatly benefit from your suggestions. Hereafter the list of your comments is reported, followed by our response. We will also provide a version of the manuscript with the tracked revisions.

**Referee #1 comments:** in the Introduction section, when the Authors speaking about the use of innovative technologies for the characterization and monitoring of landslide-affected areas (see from line 32 to line 38), including remote sensing techniques and radar interferometry (both terrestrial and satellite), in according to scientific literature, the authors should include some other references such as: Gullà et al., 2017; Peduto et al., 2017a; Tofani et al., 2014.

**Authors:** the proposed references were included.

**Referee #1** - in the section 3, at the line 131, the Authors speak in general about of a millimeter accuracy of the acquired data by GB-InSAR. Give more detail about the real accuracy (range values). A comparison with conventional ground monitoring techniques, was carried out? What are the differences on the accuracy also compared with the InSAR data provided by satellite sensors? It might be useful to provide a comparison whit the values included in the works of Nicodemo et al., 2016; Peduto 2017b; Casu et al., 2006) about the accuracy on the average velocities or displacements data derived by satellite radar sensors processed by InSAR or DInSAR techniques.

**Authors**: More details about the range value accuracy were given in the text (including the suggested works). References were also given in the discussion section about an automated total station working in the landslide area in during our research (see the manuscript revised version).

**Referee #1** - in the section 5 as well as in the figures 7,9 and 10, the Authors refer to incremental cumulative displacement (ICD) or monthly cumulated displacement (MCD) evaluated along the LOS direction. Why not along the real movement direction? Could be performed a data projection? Please, provide further details about this.

**Authors:** It is well known that a GB-InSAR system is able to measure only the component of the movement parallel to the LOS of the instrument. Thus the real displacement vector of the observed object can be calculated only if its direction is a priori known. This is one of the major limits of the technique. This is why usually the instrument is set with the view direction as parallel as possible to the expected deformations. The current paper was centered on the application of a monitoring system applied to a particular case study (a debris-flow affected slope in a mountainous inhabited area), which results could be shared with the involved technical personnel. Therefore we focused on easy interpretable data, while the data projection on the slope in order to obtain of the real movement direction will be the objective of a future work.

**Referee #1** - for a better understanding, an improvement of the Figures 3 and 11 is necessary. In particular, a visible legend should be provided.

**Authors:** The legend of Figure 3 and 11 were improved.

                                                                                    Kindest regards

* * *

[revised manuscript text omitted]

---

## Author Comment (AC2) · 3 Aug 2017

Referee #2: The English in which this paper is written is awkward in places, although comprehensible, and the authors should use shorter paragraphs. The title of the paper is misconceived. It is not a study of emergency management but a description of the deployment and use of a landslide monitoring and alarm system.

Authors: The manuscript was revised by a native English speaker, and the paragraphs were shortened.

Referee #2: The title of the paper is misconceived. It is not a study of emergency

management but a description of the deployment and use of a landslide monitoring and alarm system.

Authors: The title was changed as suggested into: "GB-InSAR monitoring of slope deformations in a mountainous area affected by debris flow events"

Referee #2: Line 9: Deep Seated Gravitational Slope Deformation - as it is not a common noun and adjectives, this term should not be capitalized. The same issue occurs with other terminology. Line 33: have been increasingly being recognized - adjust, please Line 79: Jurassic Line 90: please do not use contractions in formal prose. Figure 3 includes wording that will not reproduce at the page scale. it should be redrafted.

Authors: all of the further referee minor corrections were accepted, and the legend of figure 3 was improved.

Please also note the supplement to this comment:
https://www.nat-hazards-earth-syst-sci-discuss.net/nhess-2017-207/nhess-2017-207-AC2-supplement.pdf

**Supplement:**

Dear Revisors,
We would like to thank you for your encouraging comments, which we largely agree upon. We are sure that the manuscript will greatly benefit from your suggestions. Hereafter the list of your comments is reported, followed by our response. We will also provide a version of the manuscript with the tracked revisions.

**Referee #2 comments:** The English in which this paper is written is awkward in places, although comprehensible, and the authors should use shorter paragraphs. The title of the paper is misconceived. It is not a study of emergency management but a description of the deployment and use of a landslide monitoring and alarm system.

**Authors:** The manuscript was revised by a native English speaker, and the paragraphs were shortened.

**Referee #2**: The title of the paper is misconceived. It is not a study of emergency management but a description of the deployment and use of a landslide monitoring and alarm system.

**Authors:** The title was changed as suggested into:

"GB-InSAR monitoring of slope deformations in a mountainous area affected by debris flow events"

**Referee #2:** Line 9: Deep Seated Gravitational Slope Deformation - as it is not a common noun and adjectives, this term should not be capitalised. The same issue occurs with other terminology.

Line 33: have been increasingly being recognized - adjust, please

Line 79: Jurassic

Line 90: please do not use contractions in formal prose.

Figure 3 includes wording that will not reproduce at the page scale. it should be redrafted.

**Authors:** all of the further referee minor corrections were accepted, and the legend of figure 3 was improved.

                                                                                Kindest regards

* * *

[revised manuscript text omitted]

---

## Author Response (AR1)

*Dear Editor,*

*We would like to thank you for your encouraging comments. We are sure that the manuscript will greatly benefit from your suggestions. All the reviewers' suggestions and comments have been taken into consideration, and modified version of the manuscript were already uploaded. Hereafter the list of Your comments is reported, followed by our response. We will also provide a version of the manuscript with the tracked revisions of Your comments, while the referees' ones are embedded in the text.*

**Editor**: In section 4, the adopted monitoring system, the time line rationale of the Rotolon monitoring system and emergency management procedures is depicted in figure 6, but without any description in the text. To improve this section could be useful to move the text of lines 281-286 in section 4.

Moreover, a better description of the functioning of the early warning system and how the GB-InSAR is employed in it, is needed.

**Authors:** *The shift of suggested lines has been made and more information are now in the manuscript. Moreover, the Figure 6 has been improved.*

**Editor**: Then, the name of section 4 could be changed into The GB-InSAR monitoring strategy in the Rotolon early warning system.

**Authors:** *The title of section 4 has been changed following the suggestion.*

**Editor**: A description about how the thresholds for the 3 levels of warning (line 284) have been defined and which type of actions to undertake in each level of warning will be considerably improve the paper.

**Authors:** *We add a sentence to explain how the thresholds were defined and the action to undertake in each level of warning.*

**Editor**: The Discussion should be improved describing the role of the GB-InSAR in the decisional process of the Rotolon monitoring system and emergency management procedures. In particular, how the GB-InSAR monitoring influences the decision of issuing an alert? Reading lines 288-291 seems the 4 monitoring alerts issued were false alerts, because "Inspections carried out by the optical monitoring device and by means of field surveys from safe viewing points, assessed that detected accelerations did not generate significant slope failures". Following these considerations it seems the GB-InSAR needed to be used coupled with other monitoring strategies in order to avoid false alerts. Please discuss.

**Authors:** *The role of the GB-InSAR was better specified in section 4 (The GB-InSAR monitoring strategy in the Rotolon early warning system) and 6 (Discussion). With these explanations, we have also clarified the problem of "alerts" and the necessity of coupling with other instruments.*

Specific comments

**Editor**: Line 126. Please specify the full name for LOS. The first time you use an acronym the full name is needed.

Line 137. Please change "parallel to the line of sight (L.O.S.)", with "parallel to LOS". The full name of the acronym LOS has been specified in line 126.

**Authors:** *The necessary adjustments have been made.*

**Editor**: Line 282. Is Figure 5 or 6?

**Authors:** In this line, we are referring to figures 5 and 7. Therefore, the text is right.

**Editor**: Lines 288-289 outline the issuing of 4 monitoring alerts. What the authors mean for monitoring alert? Does it means the threshold exceedance of 5mm/h, i.e. alarm level? Please, explain.

**Authors:** *The sentence has been better specified, indicating also the alert class according to the thresholds considered*

[revised manuscript text omitted]